# Effect of Sex and Cross-Sex Hormone Treatment on Renal Monocarboxylate-Transporter Expression in Rats

**DOI:** 10.3390/pharmaceutics15102404

**Published:** 2023-09-29

**Authors:** Hao Wei, Annie Lee, Qing Zhang, Melanie A. Felmlee

**Affiliations:** Department of Pharmaceutics and Medicinal Chemistry, Thomas J. Long School of Pharmacy, University of the Pacific, Stockton, CA 95211, USAqzhang@pacific.edu (Q.Z.)

**Keywords:** monocarboxylate transporters, renal transporters, sex hormones, cross-sex hormones

## Abstract

Proton- and sodium-dependent monocarboxylate transporters (MCTs/SMCTs) are determinants of renal clearance through the renal reabsorption of monocarboxylate substrates. Prior studies with intact females and males, ovariectomized females and castrated males have revealed the hormonal regulation of renal monocarboxylate-transporter expression, prompting investigation into the regulatory role of individual hormones. The aim of the present study is to evaluate the effect of exogenous sex and cross-sex hormones on renal MCT1, MCT4, CD147 and SMCT1 mRNA and membrane-bound protein expression. Ovariectomized (OVX) females and castrated (CST) male Sprague Dawley rats received estrogen and/or progesterone, testosterone, or a corresponding placebo treatment for 21 days prior to kidney collection. The quantitative measurement of mRNA and membrane-protein levels were conducted using qPCR and Western blot. Quantitative analysis revealed the combination estrogen/progesterone treatment reduced membrane MCT1 and 4 expression and increased SMCT1 expression, while testosterone administration increased MCT1 membrane-protein expression. Correlation analysis indicated that plasma 17β-estradiol was negatively correlated with MCT1 and MCT4 membrane expression, while testosterone was positively correlated. In contrast, SMCT1 membrane expression was positively correlated with 17β-estradiol and progesterone concentrations. MCT1, MCT4, CD147 and SMCT1 renal expression are significantly altered in response to female and male sex hormones following sex and cross-sex hormone treatment in OVX and CST rats. Further studies are needed to understand the complex role of sex hormones, sex hormone receptors and the impact of puberty on MCT/SMCT regulation.

## 1. Introduction

Monocarboxylate transporters are the members of the solute carrier super family of proton-coupled monocarboxylate transporters (MCTs, SLC 16A family) and sodium-coupled monocarboxylate transporters (SMCTs, SLC 5A family) [1,2,3,4]. There are 14 isoforms within the MCT family (MCTs 1–14, SLC16A1–14), as well as two members of the sodium-dependent MCT family (SMCTs 1/2, SLC5A8/12) [2]. Monocarboxylate transporters function in the transport of short-chain monocarboxylates, including gamma-hydryoxybutyric acid (GHB), lactate, pyruvate, hormones, nutrients, and amino acids [1,2,4]. MCTs/SMCTs are involved in the influx and efflux of substrates in multiple tissues including the liver, kidney, intestine, and blood–brain barrier [1,2], which play a vital role in drug disposition and pharmacokinetics. The kidney plays an important role in maintaining body homeostasis through the urinary excretion of endogenous and exogenous compounds [5].

Proton- (MCT) and sodium (SMCT)-dependent monocarboxylate transporters are expressed in the proximal renal tubules [5,6,7]. They are involved in the renal reabsorption of GHB, and the inhibition of MCTs/SMCTs has been investigated as a therapeutic approach to treat GHB overdose [1,2,8]. In the kidney, MCT1 and MCT4 are expressed on the basolateral side of kidney proximal tubule epithelial cells [9,10]. SMCT1 is expressed in the apical membrane of tubular epithelial cells and the expression is restricted to S2–S3 segments of the proximal tubule [11]. MCTs are dependent on ancillary proteins in order to be properly trafficked to the plasma membrane [2,10,12]. The ancillary protein, CD147, has been shown to interact with MCT1, 2, 4, 11 and 12. Further, CD147 has been identified as serving as a chaperone to assist MCT1 and MCT4 in folding, stability, membrane expression, and functionality [12,13].

There are limited data on sex differences and the sex hormone regulation of MCT/SMCT expression. Previously, Enoki et al. [14] demonstrated that the administration of testosterone increased MCT1 and MCT4 protein expression in rat skeletal muscle, but their expression was not altered in the heart, which indicated that the sex hormone regulation of MCTs occurs in a tissue-specific manner. Sex differences in MCT expression have also been observed in the brain. Ovariectomy (OVX) leads to a decrease in MCT2 expression in mouse brain, suggesting that female sex hormones are also involved in MCT regulation [15]. Additionally, there was decreased MCT4 mRNA levels in 17 beta-estradiol- or dihydrotestosterone-treated rat Sertoli cells, while testosterone treatment led to a reduction in MCT2 mRNA expression [16]. Hypophysectomy resulted in an increase in testicular MCT2 mRNA levels [17]. Conversely, FSH and LH administration to hypophysectomized rats reduced MCT2 mRNA levels to basal levels [17]. For SMCTs, progesterone decreased SMCT1 protein expression with no change in the mRNA level in the kidneys of OVX female mice [18,19]. In contrast, orchiectomized mice treated with testosterone had elevated SMCT1 mRNA and protein expression [19,20].

There are minimal data in the literature assessing sex differences in MCT expression in drug-disposition organs. Our lab previously demonstrated that hepatic and renal MCTs/SMCTs varied between the sexes, over the estrous cycle, in the absence of male and female sex hormones [21], which indicated that MCT/SMCT expression can be altered by changes in sex hormone exposure. There are a lack of data on MCT/SMCT expression in response to exogenous male and female sex hormones in the cis- and trans-gender population. Therefore, the objective of the current study is to investigate sex and cross-sex hormone-dependent regulation of renal MCT1, MCT4, SMCT1 and CD147. The female sex hormones, 17β-estradiol and progesterone (alone or in combination), and the male sex hormone testosterone were administered, as well as the corresponding placebo controls. We hypothesize that renal MCT1, CD147, MCT4 and SMCT1 expression will be altered following sex and cross-sex hormone treatment, and estradiol, progesterone and testosterone are involved in altering the MCT/SMCT expression, with transporter-dependent changes in expression.

## 2. Materials and Methods

### 2.1. Chemicals

Progesterone-d9 and testosterone-d3 were purchased from Ceriliant (Round Rock, TX, USA). Formic acid was purchased from Fisher Scientific (Fair Lawn, NJ, USA). Ketamine, xylazine, and heparin were purchased from Patterson Veterinary (Saint Paul, MN, USA). High-performance liquid chromatography (HPLC)-grade acetonitrile, methanol, and acetic acid were purchased from Fisher Scientific (Fair Lawn, NJ, USA).

### 2.2. Animals, Hormone Treatment and Tissue Collection

Male and female Sprague Dawley (SD) rats, ovariectomized females and castrated males were obtained from Charles River (Wilmington, MA, USA) at 8 weeks of age. All the animals were housed in a temperature-controlled room with a 12 h day/night lighting cycle and were housed individually after surgical procedures. Standard rat chow and water ad libitum were supplied. All experiments were approved by the Institutional Animal Care and Use Committee at the University of the Pacific.

At 10 weeks of age, OVX female and CST male SD rats were implanted with testosterone, estrogen, progesterone or estrogen/progesterone combination or their corresponding placebo controls. Rats were anesthetized using isoflurane anesthesia (3–4% induction and 2% maintenance) and 60-day release pellets containing 1.5 mg 17β-estradiol and/or 50 mg progesterone, or 7.5 mg testosterone, or placebo controls (Innovative Research of America, Sarasota, FL, USA) were implanted subcutaneously in the left shoulder blade. Age-matched intact males (MC) and intact females (FC) were used as control groups. FC rats were utilized when they were in the estrus stage. Estrus cycle stages of female rats were monitored daily using vaginal lavage smear as described previously [21].

After 21 days of hormone treatment, rats were exsanguinated under isoflurane anesthesia; kidney samples were collected, and immediately snap-frozen in liquid nitrogen and stored at −80 °C until analysis. Blood was collected with a heparinized syringe, and plasma was separated using centrifugation at 5000 rpm for 15 min at 4 °C and stored at −20 °C until further analysis.

### 2.3. Quantification of Estrogen, Progesterone and Testosterone in Plasma

Progesterone and testosterone plasma concentrations were quantified using a validated LC-MS assay described in a previous publication [22] with minor modifications. Progesterone and testosterone calibration samples were prepared in acetonitrile at concentrations of blank, 0.5, 1, 2, 4, 8, 10 and 12.5 ng/mL, diluted from a steroid-mix stock solution of 200 μg/mL (Restek Corporation, Bellefonte, PA, USA). Quality control samples were prepared separately at 2 and 10 ng/mL for low QC and high QC, respectively. Progesterone-d9 and testosterone-d3 were used as internal standards.

The LC/MS system included an Agilent 1200 series UPLC consisting of an online degasser, binary pump, and autosampler connected to an Agilent 6460 Triple Quad Mass Spectrometer (Agilent Technologies, Santa Clara, CA, USA). A volume of 10 µL of the sample was injected into an InfinityLab Poroshell 120 EC-C18 column (150 × 2.1 mm i.d., 2.7-µm particle size; Agilent, Santa Clara, CA, USA). Mobile phase A was acetonitrile. Mobile phase B contained 1 mM NH_4_F in double-distilled water. A gradient elution with a flow rate of 300 µL/min was used, and the total running time was 11 min with 30% mobile phase A at 0 min, a gradient to 100% A at 10 min, and a hold at 100% A for 1 min. The retention times for progesterone and testosterone were 7.3 min and 4.9 min, respectively. The mass spectrometer was operated in a positive ionization mode with multiple-reaction monitoring. Table 1 shows the QQQ mass spectrometer conditions of progesterone, testosterone and the corresponding deuterated internal standards. The inter- and intra-day accuracy and precision based on quality control samples of the analyte were 100 ± 15%. The 17β-estradiol plasma concentrations were determined using a commercial Elisa kit (Abcam, Cambridge, UK) following the manufacturer’s instruction. The concentrations of unknown samples were calculated by interpolating the values of the samples on the standard curve obtained by Four Parameter Logistic analysis using GraphPad Prism 8.4.3. The inter- and intra-day accuracy and precision based on quality-control samples of the analyte were 100 ± 20%.

### 2.4. qPCR

RNA was isolated using an Invitrogen PureLink RNA Mini Kit (Thermo Fisher Scientific, Waltham, MA, USA) from frozen kidney tissue (approximately 30 mg), following the manufacturer’s instructions. RNA concentration and purity was determined using Nanodrop (Thermo Fisher Scientific, Waltham, MA, USA). FlashGelTM RNA cassettes (Lonza, Basel, Switzerland) were used to verify the integrity of all RNA samples prior to qPCR analysis. cDNA was synthesized from 2000 ng mRNA using an iScript Reverse Transcription Supermix (BioRad, Hercules, CA, USA). Primer sequences for quantitative real-time PCR are shown in Table 2, and were validated in our previous publication [21]. Serial dilutions (10^−2^ to 10^−10^) of plasmid DNA for each gene of interest were used as standards to confirm qPCR running conditions and efficiency. Quantitative analysis of MCT1, MCT4, CD147 and SMCT1 mRNA expression was performed utilizing iTaq Universal SYBR Green Supermix on CFX96 Connect (BioRad, Hercules, CA, USA) following the manufacturer’s instructions. The house-keeping gene 18S and Alien RNA transcript (Agilent, Santa Clara, CA, USA) were utilized for data normalization.

### 2.5. Western Blot Analysis

Membrane-bound protein was extracted from kidney tissue by using a ProteoExtract Native Membrane Extraction kit (EMD Millipore, Burlington, MA, USA), following the manufacturer’s instructions. The soluble fractions were retained to verify membrane isolation by evaluating Na^+^/K^+^ ATPase expression using Western blot. Protein concentrations were measured by using a Pierce BCA Protein Assay Kit (Thermo Fisher Scientific, Waltham, MA, USA), with bovine serum albumin as the standard.

Samples were diluted with 2× or 4× Laemmli sample buffer (BioRad, Hercules, CA, USA) containing 2-Mercaptoethanol (βME), and were heated at 37 °C for 30 min. Samples (5 μg) were separated at 200 V on a TGX Stain-Free FastCast Acrylamide Kit 10% (BioRad, Hercules, CA, USA) and transferred to Odyssey Nitrocellulose Membranes (LI-COR, Lincoln, NE, USA) at 100 V for 25 min. Membranes were incubated with 5% milk in PBST for 1 or 1.5 h at room temperature. Membranes were incubated with primary antibody anti-MCT1 (AB3540P, Millipore-Sigma, St. Louis, MO, USA) at 1:1000 dilution, anti-MCT4 (OASG04421, AVIVA, San Diego, CA, USA) at 1:2000 dilution, anti-CD147 BSG antibody (OAAJ01996, AVIVA, San Diego, CA, USA) at 1:1000 dilution, anti-SMCT1 (ARP44110_P050, AVIVA, San Diego, CA) at 1:1000 dilution or ATP1A1/ATP1A2/ATP1A3 Antibody (H-3) (sc-48345, Santa Cruz, Dallas, TX, USA) at 1:1000 dilution, with 1% milk in PBST on a shaker overnight (about 15 h) at 4 °C. The membranes were washed 3 times for 10 min in PBST and then incubated with secondary antibody, goat anti-rabbit IgG H&L(HRP) (ab97051, Abcam, Cambridge, UK) at 1:10000 dilution or peroxidase AffiniPure goat anti-rabbit IgG (111-035-144, Jackson ImmunoResearch, West Grove, PA, USA) at 1:5000 dilution, for 1 h at room temperature. The membranes were then washed for 10 min, 3 times. MCT1, MCT4 and CD147 blots were incubated with Clarity ECL reagent (BioRad, Hercules, CA, USA) for 2 min and SMCT1 blots were incubated with Radiance Plus Femtogram HRP substrate (Azure, Dublin, CA, USA) for 3 min. The bands were visualized using a ChemidocTM Touch Imaging System (BioRad, Hercules, CA, USA) and the band density was evaluated using BioRad Image Lab 6.0.1 software. The band density of each sample was normalized to the total lane density.

### 2.6. Data Analysis

Data are presented as mean ± standard deviation, and individual sample data points. Data were analyzed in GraphPad Prism 8.4.3 (Boston, MA, USA) using a one-way analysis of variance (ANOVA) with a Tukey post hoc test. A *p* value less than 0.05 was considered statistically significant, and *p* values were corrected for multiple comparisons in GraphPad Prism 8.4.3 (Boston, MA, USA). Expression of mRNA was normalized to the mean of the house-keeping genes, 18S and alien RNA, and fold-change was determined using the ΔΔC_T_ method. For analysis of membrane-protein expression, the band density of each sample was normalized to the total lane density and the band density of the inter-gel control sample. Pearson correlation analysis in GraphPad Prism 8.4.3 (Boston, MA, USA) was used to evaluate the relationships between plasma sex hormone concentrations and the membrane-protein expression of MCT1, MCT4, SMCT1 and CD147.

## 3. Results

### 3.1. Sex Hormone Concentrations

The plasma concentrations of sex hormones are presented in Table 3. Plasma progesterone concentrations were higher in the rats treated with progesterone and intact females. Plasma concentrations of 17β-estradiol were higher in the animals treated with 17β-estradiol compared to the placebo animals. A large inter-animal variation was observed in the four estrogen-treated groups, suggesting variability in estrogen disposition. Intact females had plasma 17β-estradiol concentrations of 20.97 ± 13.38 pg/mL, while intact males had plasma concentrations of 17.20 ± 5.16 pg/mL, which are consistent with the 17β-estradiol concentrations in estrus female rats and male rats from previous studies [23,24,25].

The OVX and CST groups treated with testosterone had plasma testosterone concentrations of 10.29 ± 1.206 ng/mL and 7.505 ± 1.348 ng/mL, while placebo groups had undetectable plasma testosterone levels. One OVX testosterone animal had no measurable testosterone concentration, potentially due to an issue with the pellet implantation; this animal was removed from data analysis. Intact males had a mean plasma testosterone concentration of 2.099 ± 0.404 ng/mL, consistent with previous reports [25]. One intact male had undetectable levels of testosterone, which may have been due to natural fluctuations in circulating testosterone in male rats, reported previously [26,27]. Testosterone was not detected in any female rats.

### 3.2. MCT1 Expression

#### 3.2.1. mRNA Expression

MCT1 mRNA expression in the kidney varied significantly in OVX and CST rats in response to female sex hormone treatment (Figure 1A: *p* < 0.0001, Figure 1B: *p* < 0.0001, Figure 1C: *p* = 0.0335, Figure 1D: *p* = 0.0443). OVX estrogen and OVX estrogen placebo females had significantly lower MCT1 mRNA expression as compared to males (Figure 1A). OVX estrogen rats had significantly lower expression compared to intact females. OVX progesterone rats had significantly lower MCT1 mRNA expression than placebo control rats. With cross-sex hormone treatment in CST rats, MCT1 mRNA expression was significantly lower in CST estrogen and CST estrogen placebo when compared to intact female and male animals. There were no significant differences following sex or cross-sex hormone replacement with the combination of female sex hormones. No significant differences in MCT1 mRNA expression were found between the OVX testosterone groups and intact controls (Figure 1G: *p* = 0.2552) and between the CST testosterone groups and intact controls (Figure 1H: *p* = 0.5826).

#### 3.2.2. Membrane-Bound Protein Expression

MCT1 renal membrane-protein expression following sex and cross-sex hormone treatment in OVX and CST rats varied significantly between the groups and is presented in Figure 2. (Figure 2E: *p* = 0.0400, Figure 2F: *p* = 0.0447, Figure 2G: *p* = 0.0045, Figure 2H: *p* = 0.0004). In the OVX groups, the OVX estrogen placebo and OVX estrogen animals had lower mean MCT1 membrane expression compared to the intact females and males (*p* = 0.0653). Following the combo female sex hormone treatment, the lowest MCT1 membrane expression was observed in the OVX combo rats, which was significantly lower than the OVX combo placebo group. Following cross-sex hormone treatment, the CST estrogen rats had the lowest MCT1 membrane expression, which was significantly lower than the intact males (Figure 2B: *p* = 0.0307). MCT1 expression was significantly greater in the OVX testosterone group compared to the OVX testosterone placebo (Figure 2G: *p* = 0.0179) and intact males (*p* = 0.0107). The CST testosterone group trended higher than the CST placebo group and had significantly greater expression than intact males (Figure 2H: *p* = 0.0042). Significantly greater expression was observed in the CST testosterone (*p* = 0.0010) and the corresponding placebo group (*p* = 0.0150) compared to the intact females.

### 3.3. MCT4 Expression

#### 3.3.1. mRNA Expression

The MCT4 mRNA expression profiles following sex and cross-sex hormone treatment are presented in Figure 3. MCT4 mRNA expression was significantly different following female sex hormone treatment (Figure 3C: *p* = 0.0144). The OVX progesterone placebo had the highest MCT4 mRNA expression, which was significantly higher than that of intact females and males. The OVX progesterone rats had a similar MCT4 membrane expression to that of the intact females. In the CST groups treated with female sex hormones, no significant differences were observed in response to the estrogen, progesterone and combo treatment. No significant differences in expression were found between the OVX testosterone groups and intact controls (Figure 3G: *p* = 0.7238) and between the CST testosterone groups and intact controls (Figure 3H: *p* = 0.1571).

#### 3.3.2. Membrane-Bound Protein Expression

There were significant differences in the MCT4 renal membrane-protein expression between the groups (Figure 4A: *p* = 0.0199, Figure 4B: *p* = 0.0399, Figure 4C: *p* = 0.0282, Figure 4D: *p* = 0.0015, Figure 4F: *p* = 0.0152, Figure 4H: *p* < 0.0001). In the OVX groups, the OVX estrogen and OVX progesterone placebo groups had significantly lower MCT4 membrane expression compared to intact males. In the CST animals, MCT4 membrane expression in the CST estrogen, CST progesterone, and CST progesterone placebo were significantly lower than that of the intact males. The CST progesterone group had significantly lower MCT4 membrane expression compared to that of the intact females. Following combination treatment with female sex hormones, MCT4 membrane-protein expression in the CST combo animals was significantly lower than those of the CST placebo group and the intact males. No significant difference in membrane MCT4 expression was observed between the OVX testosterone rats and intact controls (Figure 4G: *p* = 0.2161). MCT4 expression was similar between the CST testosterone and CST placebo groups. The CST testosterone and corresponding placebo rats expressed significantly higher MCT4 expression than those of the intact males (Figure 4H: *p* < 0.0001) and females (*p* < 0.0001).

### 3.4. CD147 Expression

#### 3.4.1. mRNA Expression

CD147 expression was significantly different in both the sex and cross-sex hormone-treated OVX and CST rats (Figure 5A: *p* = 0.0023, Figure 5B: *p* = 0.001, Figure 5C: *p* = 0.0046, Figure 5F: *p* = 0.0081, Figure 5G: *p* = 0.0006, and Figure 5H: *p* = 0.0300). In the OVX estrogen rats, CD147 mRNA expression was significantly decreased compared with those of the intact females and males. The OVX estrogen placebo rats had significantly lower CD147 mRNA expression than females. CD147 mRNA expression was significantly decreased in the OVX progesterone rats compared to the OVX progesterone placebo group, while the OVX progesterone placebo group had significantly higher CD147 mRNA expression compared to those of the intact females and males. For cross-sex hormone treatment in CST rats, the CST estrogen placebo group demonstrated significantly lower CD147 mRNA expression than the intact female and males. The CST combo rats had the highest CD147 mRNA expression, which was significantly higher than those of CST combo placebo group, and the intact female and males. CD147 mRNA expression was significantly greater in OVX testosterone rats compared to intact males (Figure 5G: *p* = 0.0028) and females (*p* = 0.0052). No significant difference was observed in the other groups.

#### 3.4.2. Membrane-Bound Protein Expression

CD147 membrane-protein was significantly different following the treatment with female and male sex hormones (Figure 6B: *p* = 0.0374, Figure 6G: *p* = 0.0002, Figure 6H: *p* = 0.0069). No significant differences were observed between the OVX groups following female sex hormone treatment. For cross-sex-treated CST animals, the lowest CD147 membrane-protein expression was observed in the CST estrogen rats, which was significantly lower than that of the intact male rats. The OVX testosterone rats expressed significantly greater CD147 membrane-bound protein compared to the OVX testosterone placebo group (Figure 6G: *p* = 0.0470) and the intact males (*p* = 0.0007). The CST testosterone and placebo control males expressed similar CD147 levels (Figure 6H). Both groups had significantly greater expression compared to intact females and trended higher than intact males.

### 3.5. SMCT1 Expression

#### 3.5.1. mRNA Expression

The SMCT1 mRNA expression demonstrated significant differences between sexes, and following sex and cross-sex hormone treatment in the OVX and CST animals (Figure 7A: *p* = 0.0134, Figure 7B: *p* = 0.0004, Figure 7C: *p* = 0.005, Figure 7D: *p* = 0.0006, Figure 7E: *p* = 0.1204, Figure 7F: *p* < 0.0001, Figure 7G: *p* = 0.0005, Figure 7H: *p* = 0.0008). SMCT1 had the highest mRNA expression in intact females, and was significantly higher than in intact males. SMCT1 mRNA expression in the estrogen placebo, OVX progesterone placebo and OVX progesterone groups was significantly decreased compared with that of intact females, while the OVX estrogen rats had similar SMCT1 mRNA expression levels to that of intact females. Following cross-sex hormone replacement in CST rats, SMCT1 mRNA expression in the CST estrogen placebo, CST progesterone placebo, CST progesterone and CST combo placebo groups was significantly decreased compared to intact females. Additionally, the CST estrogen and CST combo groups had significantly higher SMCT1 mRNA expression compared to the corresponding placebo-treated CST animals. The CST combo rats also had significantly higher SMCT1 mRNA expression than the intact males. A similar pattern of expression was observed in testosterone-treated OVX and CST groups compared to the intact controls (Figure 7G,H). No significant difference was found among the testosterone, testosterone placebo and intact-male groups. The testosterone and corresponding placebo males had significantly lower SMCT1 mRNA expression levels compared to intact females.

#### 3.5.2. Membrane-Bound Protein Expression

SMCT1 renal membrane-protein expression varied significantly between the sex and cross-sex hormone treatments, and the results are presented in Figure 8 (Figure 8A: *p* = 0.0414, Figure 8B: *p* = 0.0002, Figure 8C: *p* = 0.005, Figure 8D: *p* = 0.0003, Figure 8F: *p* < 0.0001, Figure 8G: *p* = 0.0008, Figure 8H: *p* < 0.0001). SMCT1 membrane expression in the OVX estrogen placebo, OVX progesterone placebo and OVX progesterone groups was significantly lower than in the intact females. Additionally, following cross-sex hormone treatment, the CST estrogen placebo, CST progesterone placebo and CST progesterone groups had significantly lower SMCT1 membrane expression compared to the intact females. In the CST estrogen and CST combo rats, SMCT1 membrane expression was significantly increased compared to the corresponding placebo groups and the intact males. No significant difference in SMCT1 membrane expression was observed between the OVX testosterone and OVX testosterone placebo groups (Figure 8G: *p* = 0.1704). The OVX testosterone rats had greater membrane expression compared to the intact males (*p* < 0.0001). The CST testosterone males had significantly lower expression levels compared to the CST testosterone placebo rats (Figure 8H: *p* = 0.008). The CST testosterone placebo rats had greater expression than the intact males (Figure 8H: *p* < 0.0001).

### 3.6. Correlation Analysis

MCT1 membrane expression was negatively correlated with plasma 17-β estradiol concentrations (r = −0.3707, *p* = 0.026) and positively correlated with plasma testosterone concentrations (r = 0.5228, *p* = 0.0013). Similarly, CD147 was negatively correlated with 17-β estradiol levels (r = −0.3908, *p* = 0.0184) and positively correlated with testosterone levels (r = 0.5056, *p* = 0.0019). MCT4 membrane expression was negatively correlated with both plasma 17-β estradiol (r = −0.3592, *p* = 0.0314) and testosterone (r = −0.4690, *p* = 0.0045) levels. No significant correlations were observed between plasma progesterone concentrations and membrane-bound MCT1, MCT4 or CD147. SMCT1 membrane expression was positively correlated with the female sex hormones estrogen (r = 0.5352, *p* = 0.0008) and progesterone (r = 0.3678, *p* = 0.0273).

## 4. Discussion

MCTs and SMCTs are determinants of drug disposition, with variations in renal-transporter expression impacting the renal clearance of transporter substrates. Our laboratory previously demonstrated differences in MCTs and SMCTs in the liver, kidney and intestine between sexes, over the estrous cycle in female rats, and in the absence of female and male sex hormones [19,21,28]. The present study explored renal MCT and SMCT mRNA and membrane-protein expression in response to exogenous the female sex hormones, estrogen and progesterone, and the androgen testosterone. This is the first study evaluating MCT and SMCT regulation in response to cross-sex hormone treatment. We have demonstrated that estrogen, progesterone and testosterone are all involved in the regulation of renal MCT1, MCT4, CD147 and SMCT1 mRNA and membrane-protein expression following sex and cross-sex hormone treatment in OVX and CST rats.

MCT1 mRNA and membrane-protein expression were significantly altered in response to female sex hormones following sex and cross-sex hormone treatment. Progesterone-treated OVX and CST rats had lower MCT1 mRNA expression (Figure 1C,D) and similar protein expression compared to corresponding placebo-treated OVX and CST rats, suggesting that progesterone may downregulate MCT1 mRNA expression. The CST estrogen, OVX combo and CST combo rats had lower MCT1 membrane expression levels compared to corresponding placebo rats (Figure 2B,E,F). These results suggest that 17β-estradiol is the driver for the observed reductions in MCT1 membrane expression. This is consistent with hepatic data showing OVX rats had significantly greater MCT1 expression than proestrus female rats and greater MCT1 membrane expression than all cycling females [21]. Additionally, renal MCT1 membrane expression trends higher in OVX females than proestrus females [28]. These results suggest that female sex hormones may be involved in the post-transcriptional regulation and membrane localization of MCT1 following sex and cross-sex hormone treatment, consistent with data from the literature demonstrating the female sex hormone-dependent down regulation of MCT1 [21,28]. The implication of the down regulation of MCT1 expression in response to female sex hormones is the decreased active renal reabsorption and increased renal clearance of GHB, suggesting the potential for reduced GHB toxicity in individuals exposed to female sex hormones.

MCT4 mRNA and membrane-protein expression varied significantly in response to female sex and cross-sex hormone treatment. The OVX progesterone rats had lower MCT4 mRNA expression compared to the OVX progesterone placebo group (Figure 3C), suggestive of the progesterone-mediated downregulation of MCT4 mRNA expression in OVX rats, consistent with sex differences observed in the literature [21,29]. Conversely, MCT4 mRNA expression was similar between the CST progesterone and CST progesterone placebo groups (Figure 3D), demonstrating a varied response to female sex hormones following cross-sex hormone treatment. CST combo rats had significantly lower MCT4 membrane expression compared to the CST combo placebo group (Figure 4D), consistent with the moderately strong negative correlation between plasma estrogen concentrations and membrane expression. Intact males and females had similar levels of MCT4 mRNA and membrane-protein expression (Figure 3 and Figure 4), which differed from hepatic MCT4 membrane expression trending higher in males [21]. The trend of MCT4 mRNA and membrane-protein expression was similar to that of MCT1 in the OVX and CST combo-treated rats, suggesting the regulation of MCT1 and MCT4 expression by female sex hormones may be driven by the same mechanism.

The testosterone-treated OVX rats had greater MCT1 membrane expression compared to the corresponding placebo groups, demonstrating the involvement of testosterone in MCT1 membrane localization (Figure 2G). The intact males had significantly lower MCT1 expression than the CST testosterone rats (Figure 2H), consistent with quantified plasma testosterone concentrations, and potentially indicative of a non-testosterone-mediated mechanism contributing to the downregulation of membrane localization in intact males. The results herein suggest testosterone-mediated upregulation and estrogen/progesterone-mediated downregulation. Thus, a possible explanation for the observed differences is the combined regulatory effects of hormones and the diurnal variation in testosterone levels [30,31]. The implication of increased MCT1 expression with exogenous testosterone administration is the increased active reabsorption and decreased renal clearance of MCT1 substrates. Previous toxicokinetic studies in our laboratory with 600 mg/kg γ-hydroxybutyrate iv showed that renal clearance was significantly lower in CST and intact males than proestrus females [28]. In addition to MCT1 abundance, the effect of sex hormones on carbonic anhydrases should be investigated. *Xenopus* oocytes injected with carbonic anhydrase-related proteins (CARP) VIII, X, or XI cRNA had significantly greater intracellular H+ concentration rates due to improved MCT1 activity with CARP expression [32]. No differences in the mean MCT4 mRNA and membrane-bound expression between the testosterone-treated OVX and CST groups were observed. Previous data from the literature show testosterone-mediated regulation is tissue- and isoform-specific: dihydrotestosterone-treated rat Sertoli cells had significantly less MCT4 mRNA expression compared to the control [16], while rats treated with testosterone for one week expressed significantly greater MCT4 expression in select hindlimb muscles; its expression in the heart was not altered [14]. Correlation analysis of plasma testosterone concentrations and MCT4 membrane expression revealed a significant moderate negative correlation in the testosterone groups, corresponding placebo-implanted groups and the intact controls. This suggests that interindividual variability in testosterone disposition contributes to altered MCT4 expression.

The CD147 membrane expression pattern resembled that of MCT1, possibly attributable to the ancillary role of CD147. CD147 mediates MCT1 and MCT4 localization, and an absence of CD147 expression results in MCT1 and MCT4 retention in the Golgi apparatus and endoplasmic reticulum [12,33]. A strong significant correlation between the membrane expression of MCT1 and CD147, but a weak correlation between the MCT4 and CD147 from the literature, also supported our observed results [29]. The CST combo-treated rats had lower membrane-protein expression levels of MCT1, MCT4 and CD147 compared to the intact females (Figure 2F, Figure 4F and Figure 6F). The difference between the CST combo and intact females suggests that additional factors may be involved in the regulation of MCT/SMCT expression, and this discrepancy may be a result of differences in hormone-receptor expression between the sexes [34,35,36]. An anticipated implication of sex hormone-mediated effects on CD147 is an altered monocarboxylate membrane expression and activity, and a subsequent change in the urinary drug excretion of MCT substrates.

The alterations in SMCT1 had differing patterns compared to MCT1 and MCT4, indicating that MCT and SMCT expression is regulated by different mechanisms. Following estrogen or progesterone treatment, SMCT1 mRNA expression and membrane-protein expression showed the same trend in the OVX and CST rats, suggesting the same role of estrogen and progesterone following sex and cross-sex hormone treatment (Figure 7A–D and Figure 8A–D). In contrast, the CST combo rats had significantly increased SMCT1 mRNA expression compared to the CST combo placebo group (Figure 7F), while no difference was observed between the OVX combo and OVX combo placebo groups (Figure 7E), suggesting that the coordinated effects of 17β-estradiol and progesterone differ following cross-sex hormone treatment. Takiue et al. [18] demonstrated that progesterone suppresses SMCT1 protein expression in mouse kidney, but that neither estradiol nor progesterone influenced the respective levels of mRNA. In the CST combo rats, SMCT1 mRNA and membrane expression were significantly increased. Additionally, a significant positive correlation between the plasma estrogen concentration and SMCT1 membrane expression was observed in the OVX and CST rats treated with estrogen, estrogen/progesterone or the corresponding placebo, and the intact animals. The intact females had significantly greater SMCT1 membrane expression than the CST testosterone, CST placebo and intact males (Figure 8), corroborating the role of female sex hormones in upregulating SMCT1 translation and membrane localization. The effects of SMCT1 expression on uric acid excretion following sex hormone treatment were investigated by Hosoyamada et al. [20]. Ovariectomized rats had significantly lower SMCT1 mRNA and protein expression than rats that underwent sham surgery, consistent with a higher urate excretion rate [20]. Testosterone-treated ovariectomized rats had significantly greater protein expression than ovariectomized rats, suggestive of testosterone-mediated increases in protein expression; their urinary excretion was higher despite their increased SMCT1 expression, possibly attributable to testosterone-modulated urate production [20].

One limitation of the present study is the lack of correlation between the observed changes in the mRNA and membrane-protein expression. One possible explanation for the incongruous relationship between mRNA and protein expression is post-transcriptional regulation, or the regulation of membrane trafficking. It is unclear which specific biochemical mechanisms are regulating these shifts in transcriptional and translational regulation. The miRNAs miR-29a and miR-29b have been shown to suppress MCT1 expression in a tissue-specific manner, such as in pancreatic beta islet cells [37]. The presence of miRNAs specifically binding and blocking MCT mRNA translation may explain the discrepancy between RNA and protein expression in response to sex hormones.

In the present study, the ovariectomy and castration surgery were performed at 8 weeks of age, at which time rats had reached puberty and had been exposed to sex hormones [38,39,40]. It is unknown whether the initial exposure to sex hormones modifies the regulation of monocarboxylate transporters, and further if these alterations are reversable after removing the endogenous sex hormones. Future studies should evaluate the impact of ovariectomy and castration prior to puberty, with subsequent sex and cross-sex hormone treatment. In the future, to elucidate the role of sex hormone receptors in the regulation of MCT/SMCT expression, studies should investigate the impact of the co-administration of sex hormones and sex hormone receptor antagonists. In addition, the administration of exogenous 17β-estradiol in the present study resulted in plasma estradiol concentrations that were higher than physiological levels [24,41]; therefore, future studies should evaluate MCT/SMCT regulation following lower doses of 17β-estradiol.

We have demonstrated that MCT1, MCT4, CD147 and SMCT1 renal expression are significantly altered in response to female and male sex hormones following sex and cross-sex hormone treatment in OVX and CST rats. The sex hormones involved and the underlying regulatory mechanisms vary between the transporters. Additionally, there were discrepancies between sex and cross-sex hormone replacement in the OVX and CST rats, suggesting that additional factors in males and females may be involved in the regulation of transporter capacities. Further studies are needed to understand the complex role of sex hormones and sex hormone receptors, and the impact of puberty on MCT/SMCT regulation, as well as the impact on GHB toxicokinetics and toxicity.

## Figures and Tables

**Figure 1 pharmaceutics-15-02404-f001:**
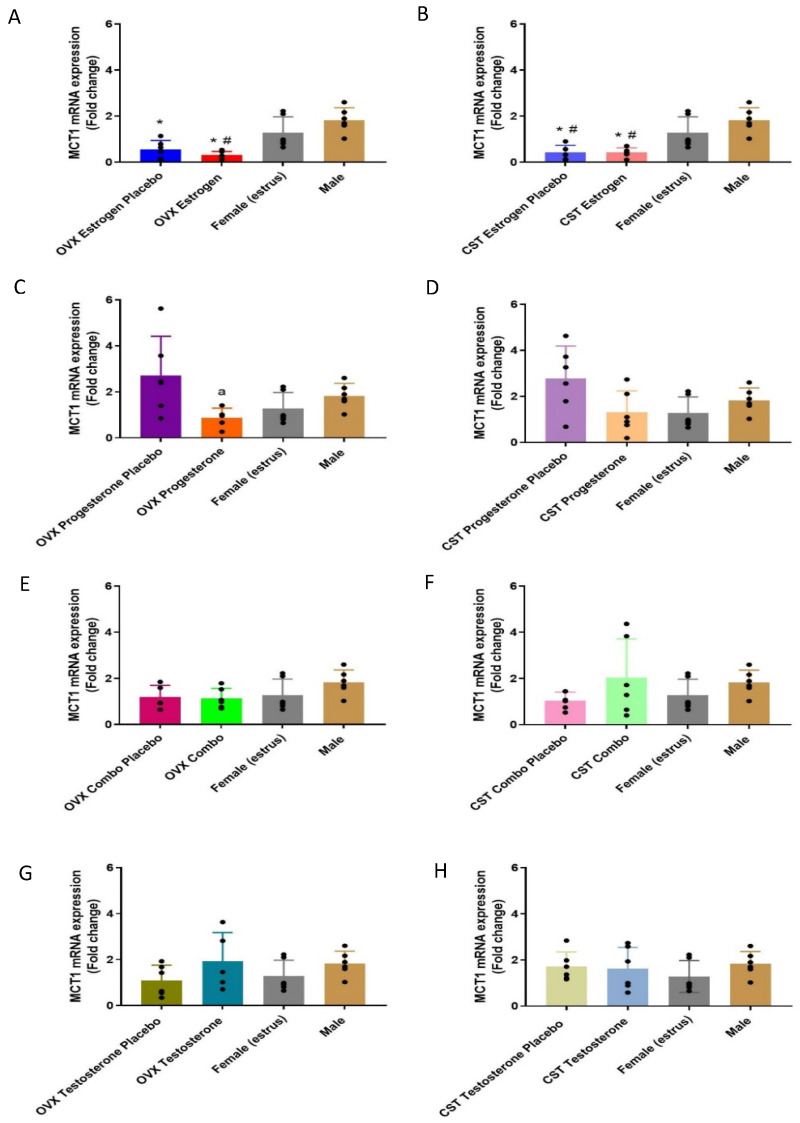
Fold-change in MCT1 renal mRNA in hormone-treated OVX CST rats. (**A**) OVX and (**B**) CST animals treated with 17β-estradiol. (**C**) OVX and (**D**) CST animals treated with progesterone. (**E**) OVX and (**F**) CST animals treated with 17 β-estradiol and progesterone. (**G**) OVX and (**H**) CST animals treated with testosterone. * *p* < 0.05 compared with males, ^#^
*p* < 0.05 compared with females (estrus), and ^a^
*p* < 0.05 compared with OVX progesterone placebo. Data are presented as mean ± SD, N = 5–6.

**Figure 2 pharmaceutics-15-02404-f002:**
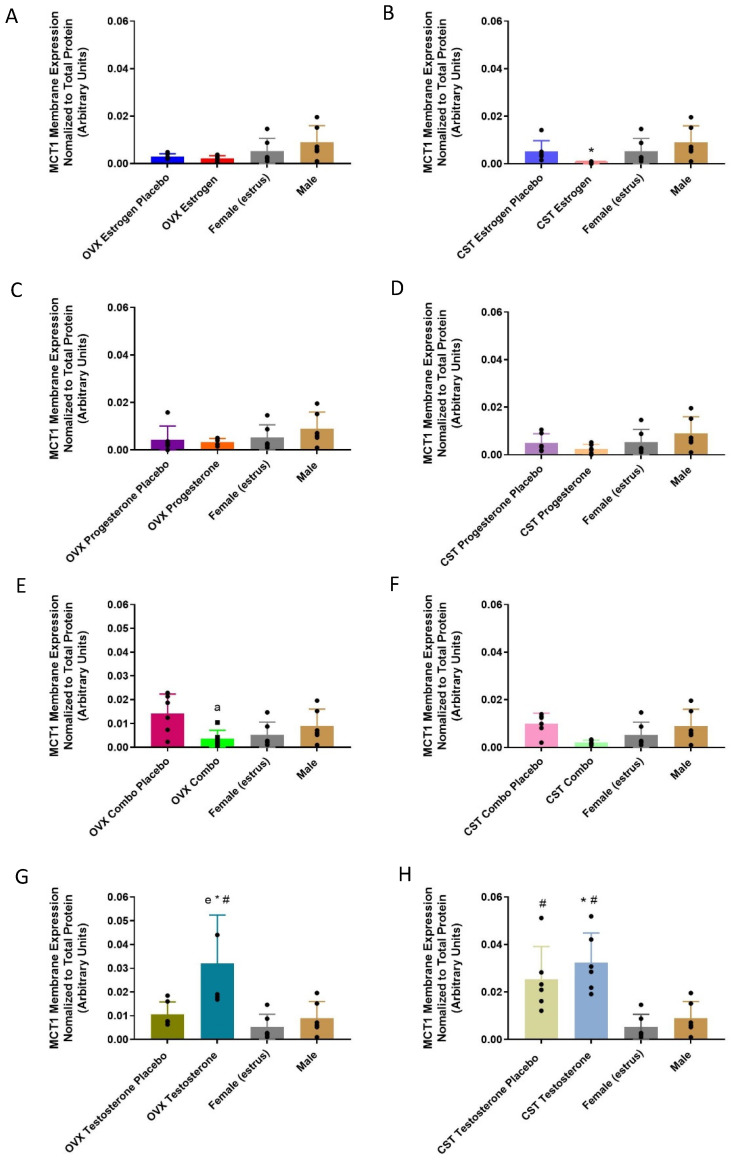
MCT1 renal membrane-bound expression in hormone-treated OVX and CST rats. (**A**) OVX and (**B**) CST animals treated with 17β-estradiol. (**C**) OVX and (**D**) CST animals treated with progesterone. (**E**) OVX and (**F**) CST animals treated with 17 β-estradiol and progesterone. (**G**) OVX and (**H**) CST animals treated with testosterone. * *p* < 0.05 compared with males, ^#^
*p* < 0.05 compared with female (estrus), ^a^
*p* < 0.05 compared with OVX combo placebo, and ^e^
*p* < 0.05 compared with OVX testosterone placebo. Data normalized to total loaded protein. Data are presented as mean ± SD, N = 5–6.

**Figure 3 pharmaceutics-15-02404-f003:**
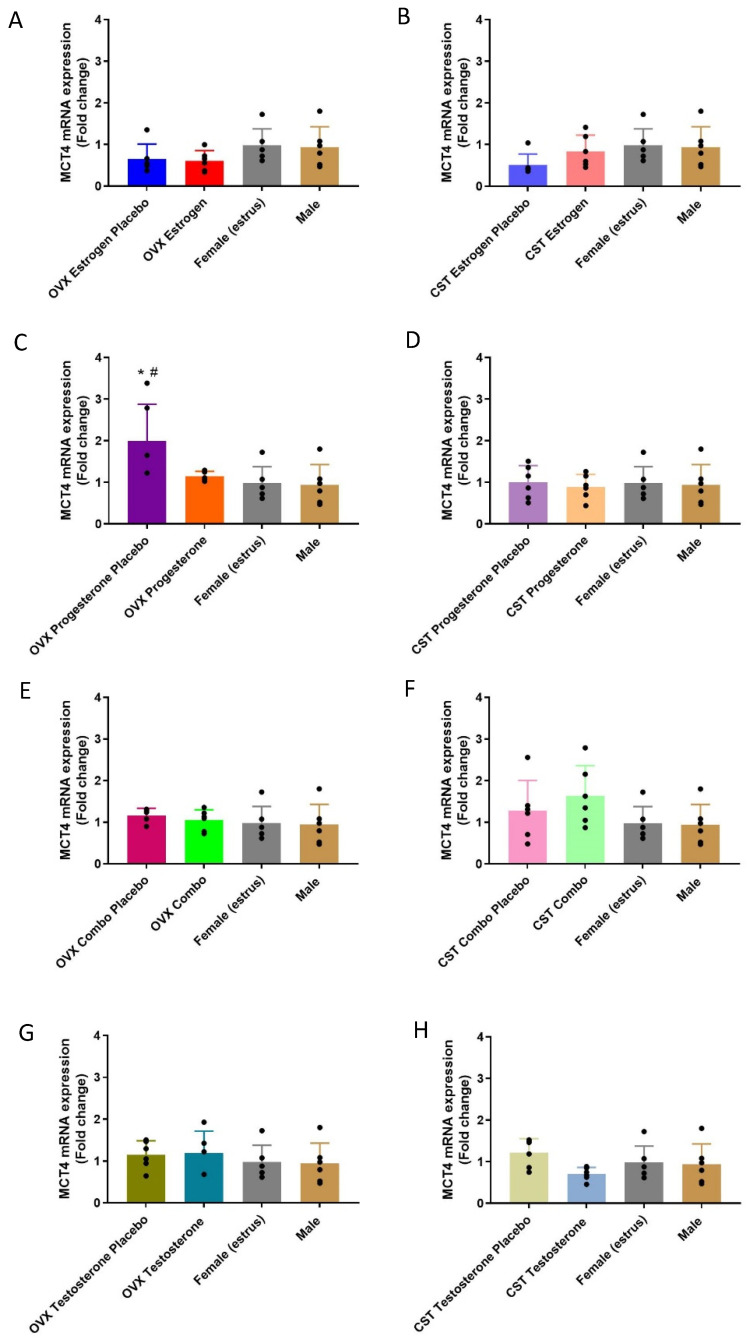
Fold-change in MCT4 renal mRNA in hormone-treated OVX and CST rats. (**A**) OVX and (**B**) CST animals treated with 17β-estradiol. (**C**) OVX and (**D**) CST animals treated with progesterone. (**E**) OVX and (**F**) CST animals treated with 17 β-estradiol and progesterone. (**G**) OVX and (**H**) CST animals treated with testosterone. * *p* < 0.05 compared with males, and ^#^
*p* < 0.05 compared with females (estrus). Data are presented as mean ± SD, N = 5–6.

**Figure 4 pharmaceutics-15-02404-f004:**
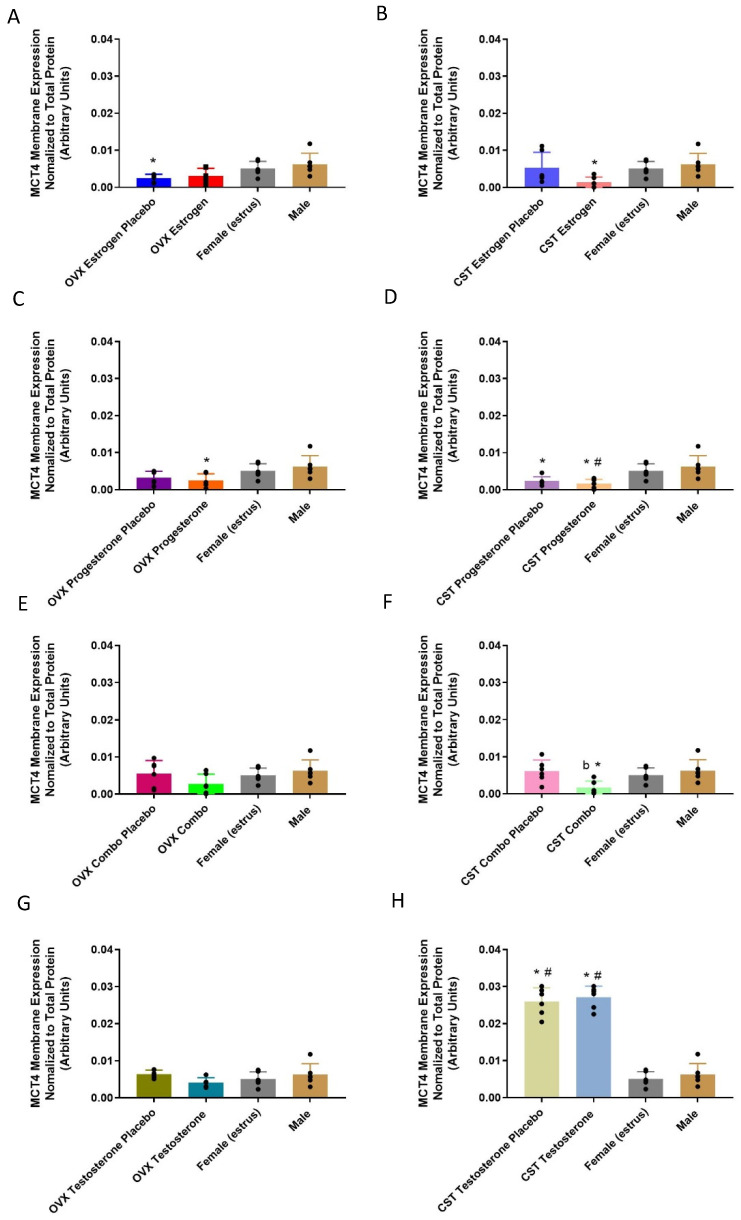
MCT4 renal membrane-bound expression in hormone-treated OVX and CST rats. (**A**) OVX and (**B**) CST animals treated with 17β-estradiol. (**C**) OVX and (**D**) CST animals treated with progesterone. (**E**) OVX and (**F**) CST animals treated with 17 β-estradiol and progesterone. (**G**) OVX and (**H**) CST animals treated with testosterone. * *p* < 0.05 compared with males, ^#^
*p* < 0.05 compared with females (estrus), and ^b^
*p* < 0.05 compared with CST combo placebo. Data normalized to total loaded protein. Data are presented as mean ± SD, N = 5–6.

**Figure 5 pharmaceutics-15-02404-f005:**
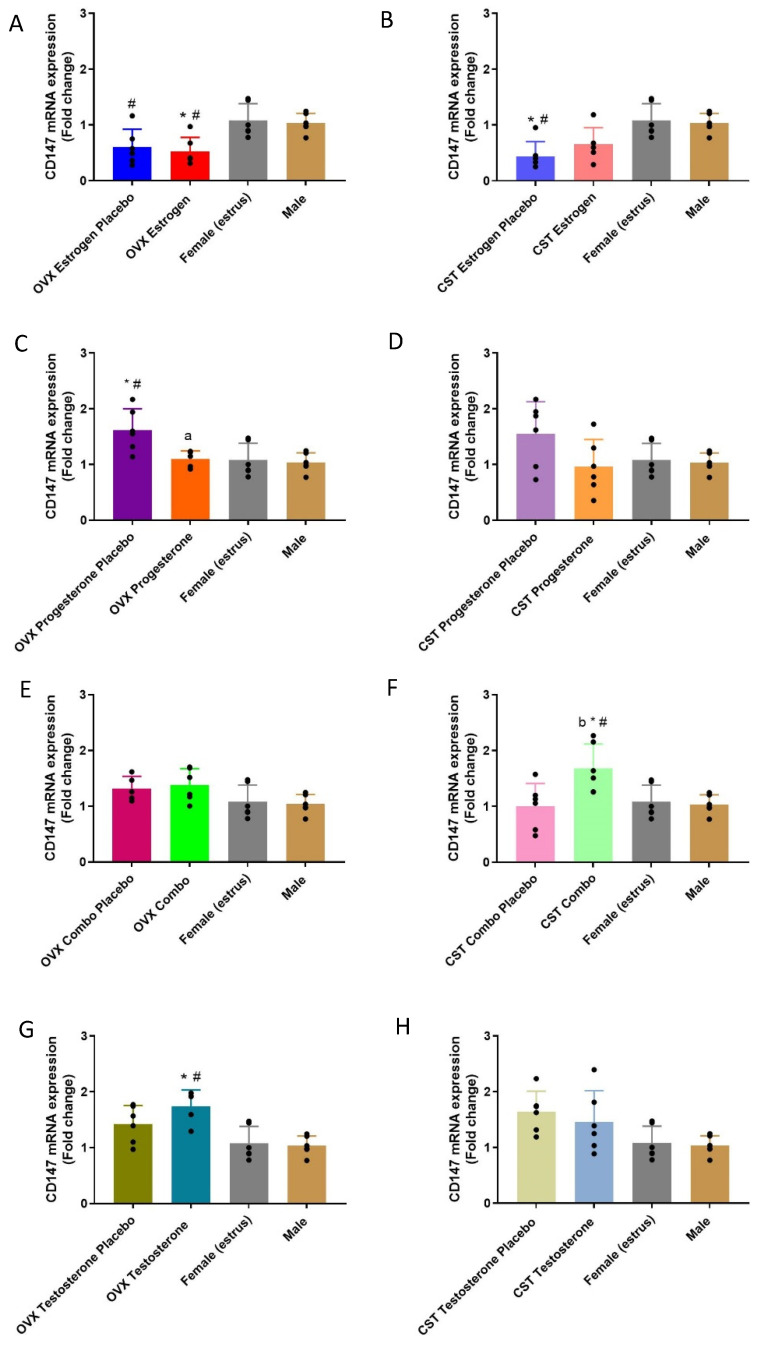
Fold-change in CD147 renal mRNA in hormone-treated OVX and CST rats. (**A**) OVX and (**B**) CST animals treated with 17β-estradiol. (**C**) OVX and (**D**) CST animals treated with progesterone. (**E**) OVX and (**F**) CST animals treated with 17 β-estradiol and progesterone. (**G**) OVX and (**H**) CST animals treated with testosterone. * *p* < 0.05 compared with males, ^#^
*p* < 0.05 compared with females (estrus), and ^a^
*p* < 0.05 compared with OVX progesterone placebo. ^b^
*p* < 0.05 compared with CST combo placebo. Data are presented as mean ± SD, N = 5–6.

**Figure 6 pharmaceutics-15-02404-f006:**
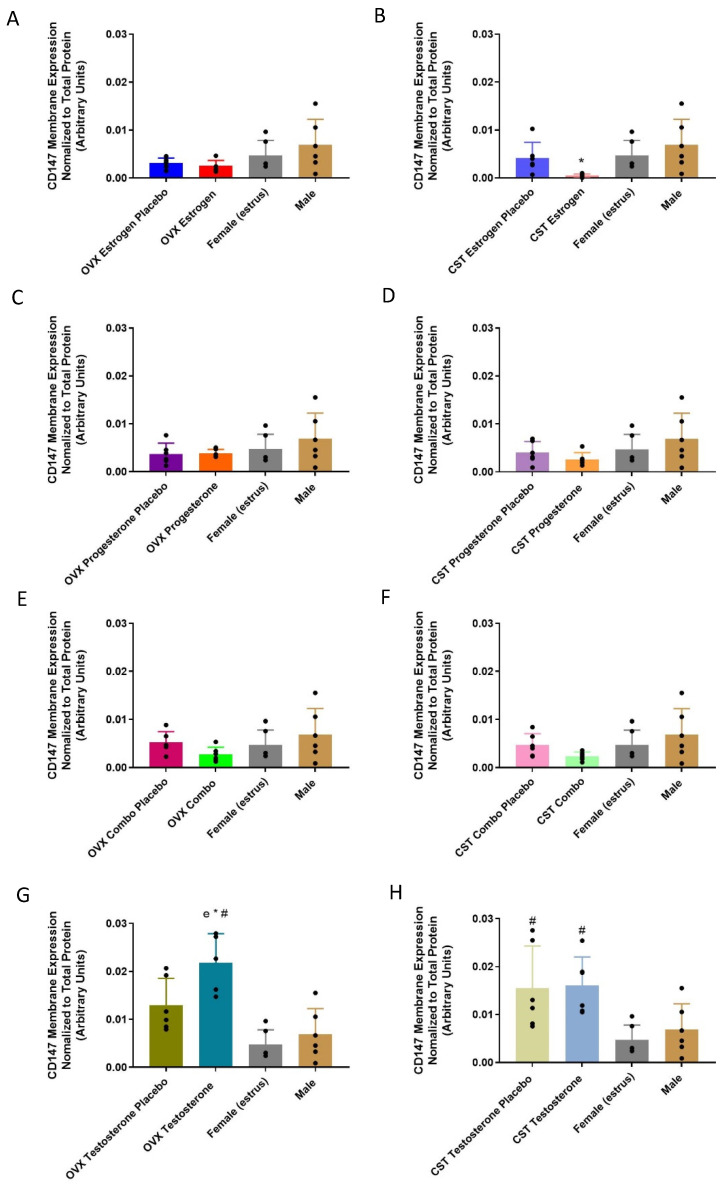
CD147 renal membrane-bound expression in hormone-treated OVX and CST rats. (**A**) OVX and (**B**) CST animals treated with 17β-estradiol. (**C**) OVX and (**D**) CST animals treated with progesterone. (**E**) OVX and (**F**) CST animals treated with 17 β-estradiol and progesterone. (**G**) OVX and (**H**) CST animals treated with testosterone. * *p* < 0.05 compared with males, and ^#^
*p* < 0.05 compared with females (estrus). ^e^
*p* < 0.05 compared with OVX testosterone placebo. Data normalized to total loaded protein. Data are presented as mean ± SD, N = 5–6.

**Figure 7 pharmaceutics-15-02404-f007:**
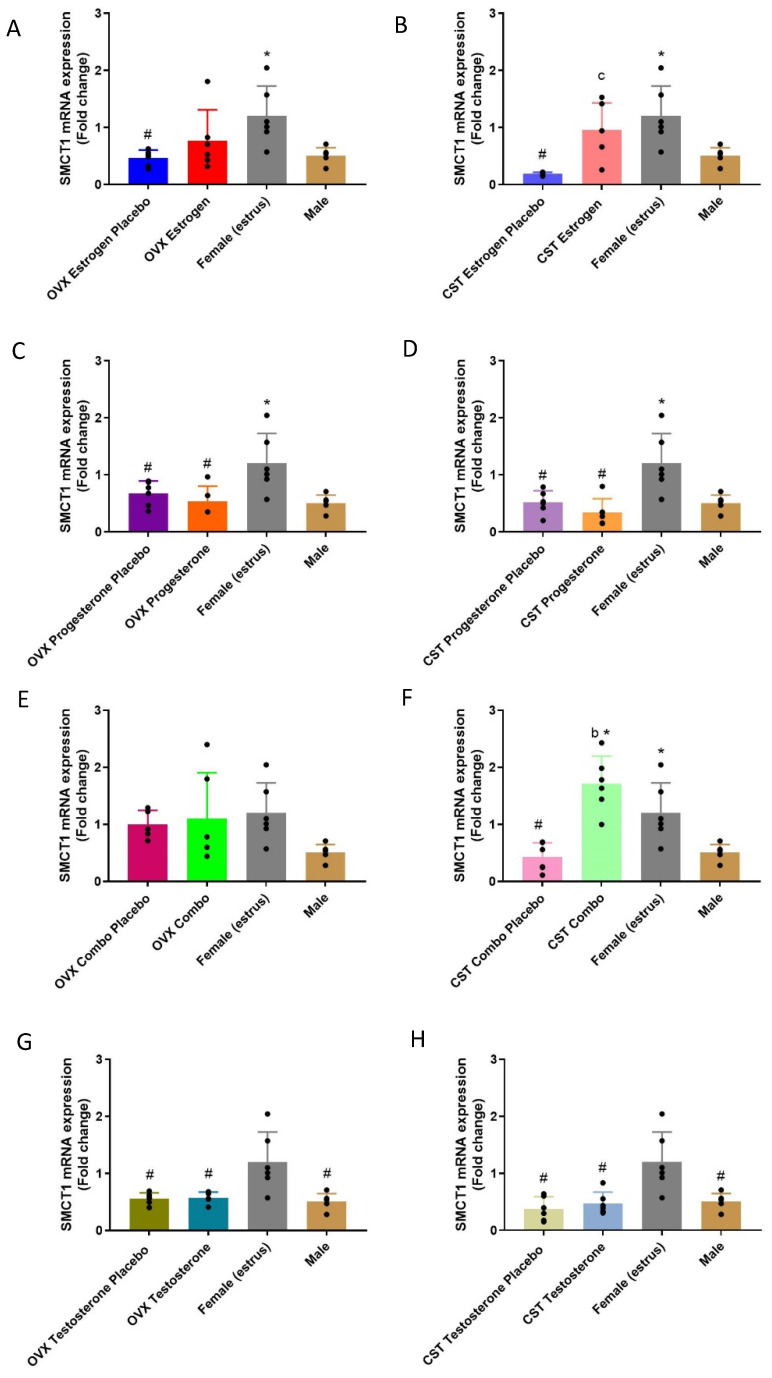
Fold-change in SMCT1 renal mRNA in hormone-treated OVX and CST rats. (**A**) OVX and (**B**) CST animals treated with 17β-estradiol. (**C**) OVX and (**D**) CST animals treated with progesterone. (**E**) OVX and (**F**) CST animals treated with 17 β-estradiol and progesterone. (**G**) OVX and (**H**) CST animals treated with testosterone. * *p* < 0.05 compared with males, ^#^
*p* < 0.05 compared with females (estrus), ^b^
*p* < 0.05 compared with CST combo placebo, and ^c^
*p* < 0.05 compared with CST estrogen placebo. Data are presented as mean ± SD, N = 5–6.

**Figure 8 pharmaceutics-15-02404-f008:**
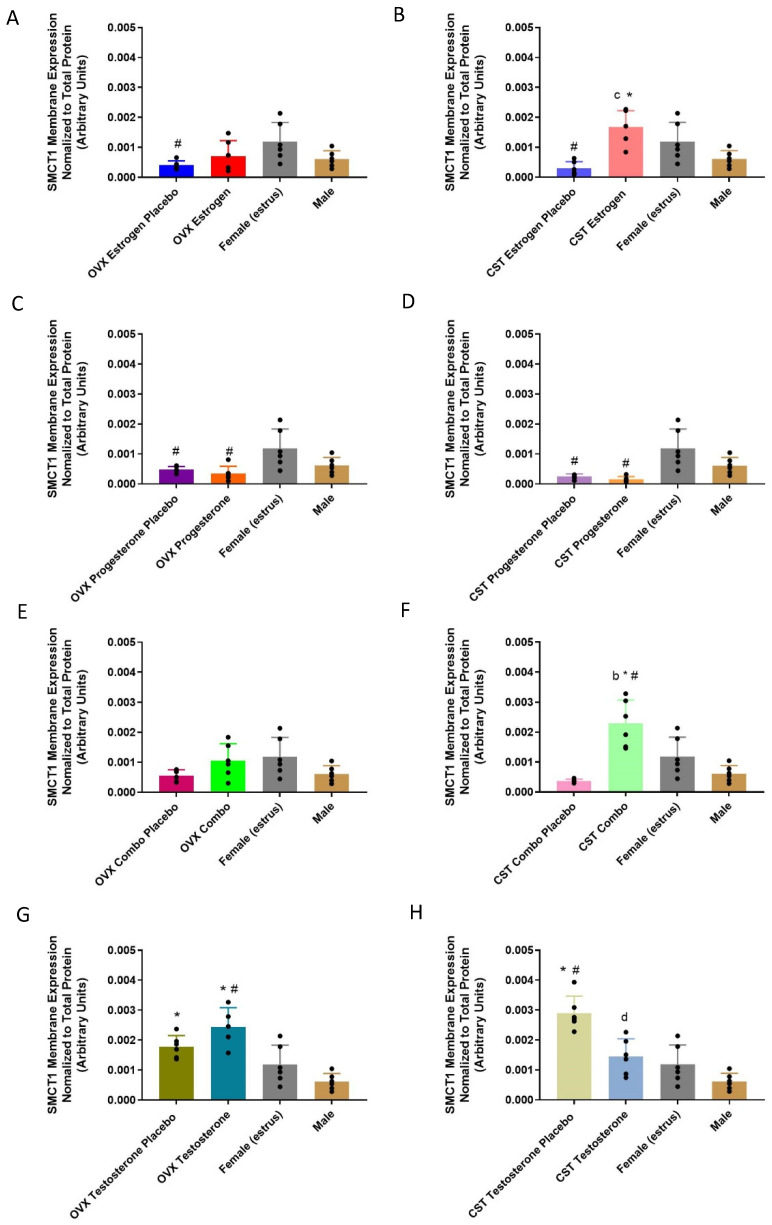
SMCT1 renal membrane-bound expression in sex hormone-treated OVX and cross-sex hormone-treated CST rats. (**A**) OVX and (**B**) CST animals treated with 17β-estradiol. (**C**) OVX and (**D**) CST animals treated with progesterone. (**E**) OVX and (**F**) CST animals treated with 17 β-estradiol and progesterone. (**G**) OVX and (**H**) CST animals treated with testosterone. * *p* < 0.05 compared with males, ^#^
*p* < 0.05 compared with females (estrus), ^b^
*p* < 0.05 compared with CST combo placebo, ^c^
*p* < 0.05 compared with CST estrogen placebo, and ^d^
*p* < 0.05 compared with CST testosterone placebo. Data normalized to total loaded protein. Data are presented as mean ± SD, N = 5–6.

**Table 1 pharmaceutics-15-02404-t001:** Mass spectrometer parameters of Triple Quad for MRM analysis of progesterone and testosterone.

Parameters	Progesterone	Progesterone-d_9_	Testosterone	Testosterone-d_3_
Q1/Q3 (+)	315.2/109.1	324.3/113.1	289.2/109.1	292.3/109.1
Fragmentor (volts)	100	117	112	107
Collision energy (volts)	26	30	26	26
Cell acceleration voltage (volts)	7	7	7	7

**Table 2 pharmaceutics-15-02404-t002:** Primer sequence and amplicon size.

Gene	Forward/Reverse Primer	Primer 5′–3′	Size (bp)
*MCT1*	Forward	GCTGTCATGTATGCCGGAG	204
	Reverse	CAATCATAGTCAGAGCTGGG	
*CD147*	Forward	GGCACCATCGTAACCTCTGT	211
	Reverse	CAGGCTCAGGAAGGAAGATG	
*MCT4*	Forward	GCTGGCTATGCTGTATGGC	185
	Reverse	TTGAGAGCCAGACCCAAGC	
*SMCT1*	Forward	GTTGCTGGTGGGGATTCTTA	200
	Reverse	CCACTGTGGTCTGGGAAGTT	
*r18s*	Forward	GTTGGTTTTCGGAACTGAGGC	206
	Reverse	GTCGGCATCGTTTATGGTCG	

**Table 3 pharmaceutics-15-02404-t003:** Female and male sex hormone levels in intact females, males, and hormone-treated OVX and CST rats.

Plasma Hormone Level	17β-Estradiol (pg/mL)	17β-Estradiol Non-Detected	Progesterone (ng/mL)	Progesterone Non-Detected	Testosterone (ng/mL)	Testosterone Non-Detected
OVX Estrogen Placebo	24.27 ± 6.674	0 out of 6	1.916 ± 1.120	1 out of 6	0	6 out of 6
OVX Estrogen	744.9 ± 406.1	0 out of 6	4.511 ± 3.841	0 out of 6	0	6 out of 6
OVX Progesterone Placebo	25.46 ± 14.63	0 out of 6	2.169 ± 2.242	2 out of 6	0	6 out of 6
OVX Progesterone	25.65 ± 11.32	0 out of 6	4.000 ± 1.673	0 out of 6	0	6 out of 6
OVX Placebo Combo	11.62 ± 7.645	0 out of 6	0.825 ± 0.284	3 out of 6	0	6 out of 6
OVX Combo	1104 ± 285.5	0 out of 6	6.241 ± 4.011	0 out of 6	0	6 out of 6
OVX Testosterone placebo	--	--	2.703 ± 2.127	2 out of 6	0	6 out of 6
OVX Testosterone	--	--	2.095 ± 1.063	1 out of 6	10.29 ± 1.206	1 out of 6
Female (Estrus)	20.97 ± 13.38	0 out of 6	8.026 ± 2.191	0 out of 6	0	6 out of 6
CST Estrogen Placebo	12.99 ± 7.311	0 out of 6	0.711 ± 0.179	2 out of 6	0	6 out of 6
CST Estrogen	1342 ± 691.2	0 out of 6	1.829 ± 0.922	1 out of 6	0	6 out of 6
CST Progesterone Placebo	24.48 ± 3.775	1 out of 6	0	6 out of 6	0	6 out of 6
CST Progesterone	12.81 ± 3.662	0 out of 6	6.815 ± 4.204	0 out of 6	0	6 out of 6
CST Placebo Combo	10.48 ± 4.896	0 out of 6	1.377 ± 0.689	1 out of 6	0	6 out of 6
CST Combo	1385 ± 846.7	0 out of 6	4.749 ± 2.423	0 out of 6	0	6 out of 6
CST Testosterone placebo	--	--	1.091 ± 1.091	3 out of 6	0	6 out of 6
CST Testosterone	--	--	1.143 ± 1.143	4 out of 6	7.505 ± 1.348	0 out of 6
Male	17.20 ± 5.161	0 out of 6	1.791 ± 1.496	3 out of 6	2.099 ± 0.404	1 out of 6

N = 5–6 in each group. Data are presented as mean ± SD of detected hormone levels in each group.

## Data Availability

The data presented in this study are available within the article and Appendix A.

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
