# Peer review of "Effect of Sex and Cross-Sex Hormone Treatment on Renal Monocarboxylate-Transporter Expression in Rats"

_pharmaceutics, 2023, doi:10.3390/pharmaceutics15102404_

Round 1

Reviewer 1 Report

In this manuscript, Wei et al. examined the effect of exogenous sex and cross-sex hormones on renal MCT1/4, CD147, and SMC4 mRNA and membrane protein expression in ovaritomized (OVX) female, castrated (CST) male, and control rats. They measured the plasma levels of 17beta-estradiol, progesterone, and testosterone in rats by using LC/MS, determined the renal mRNA levels of the 4 genes by using RT-PCR, and examined the membrane levels of corresponding proteins by Western blotting. As the regulation of renal monocarboxylate transporters by sex hormones has not been well explored, the data should be useful to the related fields. My specific comments and concerns are presented below.

Major:

1.       The authors suggested that renal monocarboxylate transporters play a vital role in drug disposition and PK. However, while the transporters might be important in maintaining multiple endogenous metabolites, their role in drug disposition and PK remain unclear or limited. The authors should provide literature to support their point in the introduction.

2.       For the plasma levels of sex hormones that they measured (Table 3), the number of significant digits was inappropriate.

3.       On many occasions, there was inconsistence between mRNA and membrane protein results. Please explain and discuss.

4.       They provided raw gels for the proteins of interest. However, the gels for the corresponding internal control were not provided, making it’s impossible to assess their results regarding the membrane expression of proteins.

5.       The data interpretation in the discussion seemed to be inconsistent with the results. For example, beginning with line 206, “OVX and CST rats treated with estrogen had lower MCT1 membrane expression, and OVX combo and CST combo rats had significantly lower MCT1 membrane expression compared to corresponding placebo rats”. However, this was only true for Figure 2E.

6.       In line 228, “while females had lower membrane protein expression than males”. In fact, no significant changes were found between females and males in Figure 3&4.

7.       In line 233, “Testosterone treated OVX and CST rats had greater MCT1 membrane expression compared to corresponding placebo groups demonstrating the involvement of testosterone in MCT1 membrane localization”. However, this was only true for OVX rats.

8.       In general, the authors selectively discussed their positive results even with which inconsistency existed between results and discussion. This could be very misleading and wrong conclusions among readers. I suggest citing Figure/Table number in appropriate places in their discussion.

Minor:

1.       In the abstract, SMCT4 should be SMCT1.

2.       Introduction, Line 44/45, please consider revising. The two sentences were essentially the same.

3.       Introduction, Line 65, revise “While”.

4.       Materials and Methods, Line 77, what are GHB, NIDA,…? Please do not assume that the readers would know your abbreviations.

5.       Table 3, 17B should be 17beta.

6.    Please carefully examine your references. For example, I could not find reference 24 in PubMed.

Acceptable; Moderate editing would be necessary.

Reviewer 2 Report

Review of Manuscript ID JPET-2023-12345: "Effect of Sex and Cross-Sex Hormone Treatment on Renal Monocarboxylate Transporter Expression in Rats"

The study investigated the regulation of renal monocarboxylate transporters (MCT1 and MCT4), CD147, and sodium-coupled monocarboxylate transporter (SMCT1) mRNA and membrane protein expression in response to exogenous female sex hormones (estrogen and progesterone) and the male sex hormone testosterone. The manuscript is well-structured, and the research addresses an important topic in pharmacology, namely the influence of sex hormones on drug disposition. However, I have identified several critical comments and suggestions that should be addressed before considering this manuscript for publication:

Lack of Clear Hypothesis Statement:

The manuscript lacks a clear and concise statement of the research hypothesis. While it is apparent that the study aimed to investigate the impact of sex hormones on renal monocarboxylate transporters, it is essential to state the research question or hypothesis at the outset explicitly. This will provide readers with a clear understanding of the study's objectives.

Discussion of Methodological Limitations:

The manuscript does not adequately discuss potential methodological limitations of the study. Given the complexity of the interplay between sex hormones and transporter regulation, the authors should provide a comprehensive discussion of potential confounding factors or limitations in the study design. Additionally, the manuscript should address any limitations related to the animal model used, sample size, and the choice of specific sex hormone treatments.

Statistical Analysis and Interpretation:

The statistical analysis section should be more detailed and transparent. The manuscript should provide additional information regarding the statistical methods used, including details on the choice of statistical tests, correction for multiple comparisons, and statistical significance thresholds. Furthermore, the interpretation of the results should be cautious, considering the potential for Type I errors when multiple comparisons are performed.

Clarity of Results:

While the results section is comprehensive, it can be challenging to follow due to the extensive data presented. The authors should consider using tables or figures to summarize key findings visually, making it easier for readers to grasp the main outcomes of the study.

Biological Significance:

The manuscript would benefit from a more in-depth discussion of the biological significance of the findings. For instance, how do the observed changes in transporter expression relate to renal function and drug disposition in rats? Providing a broader context for the results would enhance the manuscript's relevance to pharmacologists and researchers in related fields.

Recommendations for Further Research:

The study opens up intriguing questions regarding the impact of sex hormones on drug pharmacokinetics. The authors should conclude the manuscript by proposing specific directions for future research in this area, including potential clinical implications of their findings.

Minor comments:

1) The manuscript mentions that plasma estradiol concentrations were higher than physiological levels. This raises questions about the relevance of the hormone dosages used and whether they accurately represent typical hormone levels in both humans and rats.

2) The study lacks control groups that received no hormone treatment. These control groups would be valuable for understanding baseline MCT expression and how it changes in response to sex hormones.

3) The sample sizes for each treatment group are not clearly specified in the provided text. It's important to know the sample sizes for each group to assess the statistical power of the study.

4) The study mentions "large inter-animal variation," which could introduce significant variability in the results. It's essential to explore the potential causes of this variation and how it might affect the robustness of the findings.

In summary, while the manuscript presents valuable data on the regulation of renal monocarboxylate transporters and CD147 by sex hormones, addressing the above-mentioned comments will improve the clarity, robustness, and overall quality of the manuscript. 

Round 2

Reviewer 1 Report

My concerns have been mostly addressed.  

Editing of language remains to be needed.